# Does Frenotomy Modify Upper Airway Collapse in OSA Adult Patients? Case Report and Systematic Review

**DOI:** 10.3390/jcm12010201

**Published:** 2022-12-27

**Authors:** Eduardo J. Correa, Carlos O’Connor-Reina, Laura Rodríguez-Alcalá, Felipe Benjumea, Juan Carlos Casado-Morente, Peter M. Baptista, Manuele Casale, Antonio Moffa, Guillermo Plaza

**Affiliations:** 1Otorhinolaryngology Department, Hospital Quirónsalud Marbella, 29603 Málaga, Spain; 2Otorhinolaryngology Department, Clínica Universidad de Navarra, 31007 Pamplona, Spain; 3School of Medicine, Campus Bio-Medico University, Unit of Integrated Therapies in Otolaryngology, Fondazione Policlinico Universitario Campus Bio-Medico, 00128 Rome, Italy; 4Otorhinolaryngology Department, Hospital Universitario de Fuenlabrada, Universidad Rey Juan Carlos, 28042 Madrid, Spain

**Keywords:** obstructive sleep apnea, myofunctional therapy, Iowa oral performance instrument (IOPI), tongue-tie, frenulum, frenotomy

## Abstract

Ankyloglossia (tongue-tie) is a condition of the oral cavity in which an abnormally short lingual frenulum affects the tongue’s mobility. Literature on the correlation between ankyloglossia and obstructive sleep apnea (OSA) is scarce. The main objective of this study was to report our preliminary experience in adult OSA patients before and after ankyloglossia treatment, using drug-induced sleep endoscopy (DISE) to evaluate the upper airway modifications resulting after treatment, and to present a systematic review of the impact of ankyloglossia and its treatment on OSA adults. We found that, after frenotomy, regarding the DISE findings, and according to the VOTE classification, two of the three patients showed an improvement in tongue level, from 2A-P (complete anteroposterior collapse) to 1ap (partial anteroposterior collapse). The third patient showed no changes in his UA after frenotomy, neither worsening nor showing improvement. Thus, the results of this study suggest that frenotomy in OSA patients with ankyloglossia could reduce tongue collapse, probably by allowing the tongue to take into the physiological position in the oral cavity. These patients should undergo speech therapy and oropharyngeal exercises prior to any surgical procedure, in order to avoid glossoptosis and to improve the quality of life and sleep apnea results.

## 1. Introduction

Ankyloglossia—commonly named “tongue-tie”—is a condition of the oral cavity in which an abnormally short lingual frenulum affects the tongue’s mobility by tethering it to the floor of the mouth. Its incidence is estimated to be between 3–11% of the pediatric population [1,2,3,4], being slightly more common in male subjects.

Its most recent definition, by the American Academy of Otolaryngology–Head and Neck Surgery (AAO-HNS) consensus, describes it as a “condition of limited tongue mobility caused by a restrictive lingual frenulum” [5]. During fetal development, the frenulum balances the tongue, lip muscles, and growing facial bones, before its retraction after birth, while in ankyloglossia there is a failure to recede [6,7].

The clinical presentation of ankyloglossia varies from asymptomatic to adverse consequences, such as impairing the sucking, chewing, deglutition, speaking functions, and orthodontic anomalies [8,9,10]. Moreover, this condition may contribute to oromyofacial dysfunction [6] and obstructive sleep apnea (OSA) [11].

Several methods for the clinical assessment of tongue mobility and ankyloglossia have been published by Kotlow [12], Ruffoli [9], Hazelbaker [13], Marchesan [14], or Ingram [15]. During the office examination of OSA patients, we routinely follow the “Tongue+ Protocol” [16] using both the Marchesan and Hazelbaker tests [13,14]. 

Although the literature on the correlation between ankyloglossia and OSA is scarce, as shown in a systematic review conducted by Chinnadurai et al. in 2015 [17], different authors have described the incidence of palatal and maxillomandibular disorders in patients with tongue-tie [10,11]. Speech therapy and surgical procedures, such as frenotomy, are currently the treatment options for patients with ankyloglossia and clinical consequences. A recent study by Yuen et al. found an independent association between low tongue mobility and a higher risk of OSA, concluding that reduced tongue mobility is associated with the occurrence and severity of OSA [18].

OSA is treated mainly with continuous positive airway pressure (CPAP) therapy, mandibular advancement devices, or surgical procedures [19,20]. Since Guimaraes et al. [21] published their results of myofunctional therapy (MFT), this approach has been presented as a promising complementary treatment modality, supported by increasing published evidence. MFT has shown a reduction in snoring, the apnea-hypopnea index (AHI), oxygen desaturation, and daytime sleepiness [22,23,24,25]. As it can be difficult for our patients to access a professional speech therapist, we recommend them to carry out a home-based therapy with oropharyngeal exercises, using a mHealth app (Airway Gym^®^) which has demonstrated acceptable rates of adherence and results [26,27,28,29].

Given that MFT is based on oral exercises, patients with ankyloglossia are not good candidates for this modality, as the condition presents with impaired tongue mobility that will limit its success. Thus, we suggest lingual frenulum surgery (frenotomy) as an option for these patients to improve their results. Furthermore, if the tongue can relevantly participate in the upper airway (UA) collapse in OSA, when ankyloglossia is reversed, we might expect an OSA improvement. A drug-induced sleep endoscopy (DISE) is a diagnostic tool to assess the UA of snorers and OSA patients in conditions that mimic natural sleep [30]. DISE may be used to assess such improvements.

The main objective of this study was to report our preliminary experience in adult OSA patients before and after ankyloglossia treatment using DISE to evaluate the UA modifications resulting after treatment, and to present a systematic review of the impact of ankyloglossia and its treatment on OSA adults.

## 2. Materials and Methods

### 2.1. Systematic Review 

As lingual frenulum surgery has been given different names in the literature, such as frenectomy, frenotomy, frenulectomy, frenuloplasty, miofrenuloplasty, lingual frenum release, and tongue-tie surgery, we designed a systematic review using all these medical subject readings (MESH) terms and sleep apnea or snoring in adults (PRISMA-Search is provided as Appendix A).

The review followed the PRISMA guidelines [31], and the protocol was registered in PROSPERO (CRD42022360552). Databases PubMed, Cochrane, Scopus, Web of Science, Google Scholar, and ProQuest were searched (search strategy and results can be found as complementary material). Deduplication was fulfilled using SR-Accelerator^®^ [32] and the studies selection with Rayyan^®^ [33].

The inclusion criteria were: adult patients (16 years old or more); diagnosis of snoring and/or OSA; clinical diagnosis of ankyloglossia following a published and reproductible assessment tool/protocol; lingual frenulum surgery (no distinction of the surgical technique).

Exclusion criteria were: OSA diagnosis not confirmed by a standardized sleep test (as by questionnaires); syndromic patients with craniofacial malformations; neurological comorbidities with swallowing or speech impairment; other/s upper airway surgical procedure/s performed concurrently.

### 2.2. Case Report

We analyzed three cases of adult patients with a clinical diagnosis of ankyloglossia following the “Tongue+Protocol” [16] and severe OSA based on a home sleep apnea test (HSAT). Following the Marchesan protocol, patients were asked to place the tongue behind the maxillary incisors, and then open their mouth while still in contact. The first measurement of mouth opening was taken. After that, the patient was asked to open his mouth with the tongue resting down, and a second measurement was taken. A difference of 50% or more was considered a pathological lingual frenulum [14,16]. All patients were male, with no adverse nasal or maxillomandibular findings, and who had acceptable strength parameters for the tongue and buccinator measured by the Iowa Oral Performance Instrument (IOPI^®^) [34,35]. They performed MFT with the mHealth app Airway Gym^®^ and presential speech therapy for the tongue-tie, with the objective being to improve tongue mobility. After showing no improvements on MFT, patients were offered a frenotomy to provide the tongue with adequate mobility. Patients agreed and signed the informed consent regarding the surgical procedure and the permission to publish an anonymized clinical record, recordings, and results.

On the day of the scheduled frenotomy, we performed a three-step procedure. First, we applied DISE (see below), then waited for the patient to wake up spontaneously and proceeded with the second step—frenotomy under local anesthesia with a collaborative patient, using cold dissection and bipolar coagulation, if required. In the third step, we performed an immediate postoperative DISE to assess any changes in the upper airway.

The DISE protocol was performed under sedation with propofol and bispectral index scale (BIS) monitoring, and the results were registered according to the VOTE classification [36] as recommended on the European position on DISE [37,38].

Three doctors trained in OSA and DISE reviewed the anonymized DISE recordings separately, without knowing which was preoperative or postoperative, and the results were recorded.

## 3. Results

### 3.1. Systematic Review Results

After removing 95 duplicates, 240 articles were screened, of which 201 were excluded by the reading of the title or abstract. We could not retrieve three of them, resulting in 36 eligible studies. Of the 36 eligible studies (Figure 1), 29 were discarded after a full-text reading for not fulfilling the inclusion and exclusion criteria: population = 20 studies (pediatric or syndromic patients); intervention = seven (other intervention, techniques comparison, medical intervention); and outcome = two (not related to sleep apnea). Finally, we selected the seven studies included in this systematic review (Table 1).

### 3.2. Case Report Results

Baseline data were as follows:Patient number 1 was male and 54 years old, with a body mass index (BMI) of 28 kg/m^2^, an apnea/hypopnea index (AHI) of 31, maximum oxygen desaturation of 81%, presenting with a Friedman tongue position (FTP) [44] grade 2, and IOPI^®^ scores: tongue (67 kPa) and buccinator (34 kPa).Patient number 2 was male and 48 years old, with a BMI of 33 kg/m^2^, AHI 37, maximum oxygen desaturation 72%, FTP 2, and IOPI^®^ scores: tongue (59 kPa) and buccinator (28 kPa).Patient number 3 was male and 42 years old, with a BMI of 27 kg/m^2^, AHI 62, maximum oxygen desaturation of 85%, FTP 2, and IOPI^®^ scores: tongue (49 kPa) and buccinator (32 kPa).

After the frenotomy, regarding the DISE findings, there were no changes in the velum, oropharynx, or epiglottis in any case, when comparing the preoperative and immediate postoperative records. According to the VOTE classification, two of the three patients showed an improvement in their tongue level, from 2A-P (complete anteroposterior collapse) to 1ap (partial anteroposterior collapse). The third patient showed no changes in his UA after frenotomy, neither worsening nor showing improvement.

Video recordings of the DISE procedures are provided as Appendix A.

## 4. Discussion

In 2020, the AAO-HNS published the Clinical Consensus Statement on Ankyloglossia in Children [5], hypothesizing that the “anterior tethering of the tongue serves, to some degree, to prevent posterior collapse of the tongue and that if the frenulum is released, it could lead to worsening OSA”. However, when reviewing the references in the consensus, this concept is based on two case reports [45,46].

The first case report referred was published in 1995, describing that, after induction with anesthesia and frenotomy, the patient presented with signs of UA obstruction [45]. The authors did not specify whether the muscular strength was evaluated or if the patient performed speech therapy before the frenotomy. Moreover, they did not determine which signs of UA obstruction were identified (possibly oxygen desaturation, a known complication of general anesthesia). In the same report, the authors reported that “…after surgical release, the genioglossus may not be able to generate sufficient force to prevent airway collapse…” a concept that supports our theory that muscular tone needs to be improved prior to the surgical procedure. We also consider the use of local anesthetics relevant, as general anesthesia may lead to impaired genioglossus function, as described in this referred case report. We must also consider that this procedure was performed before the standardization of DISE and knowledge of the patterns of airway collapse.

The other publication referred to by the AAO-HNS was a two-case report by Genther et al. in 2015 [46], regarding patients with a Pierre Robin sequence speech defect. These patients present with severe micrognathia and glossoptosis. We agree with the authors that the high risk of UA obstruction in this syndrome is caused by glossoptosis secondary to micrognathia. This is a specific population of a rare disease, with an incidence of 1/8500 to 1/30,000 newborns [47], and we cannot extrapolate this situation to the general population with OSA.

Our systematic review has found that there are recent publications regarding the association between ankyloglossia and OSA. The recent literature review published by Bussi et al. [41], showed that an untreated shortened lingual frenulum is associated with OSA, and also alters the craniofacial growth with respiratory consequences. The authors retrieved four articles for inclusion, in accordance with the difficulties that we found in collecting studies addressing our concern. Regarding ankyloglossia surgery, they concluded that this procedure in association with MFT can improve quality of life and snoring. 

In 2016, Guilleminault et al. published a retrospective study on 150 pediatric patients [11], addressing to correlate ankyloglossia with consequent OSA, by clinical investigation and polysomnographic evaluation, concluding that ankyloglossia left untreated is associated with OSA at a later stage in life. Although it encompasses a pediatric population, the study comes to value as the authors used an objective sleep study to diagnose OSA.

The retrospective cohort published in 2019 by Zaghi et al. [39] concluded that surgical treatment with MFT is a safe procedure, and potentially effective for the treatment of snoring. Although it is a promising outcome, we must take into consideration that the study was carried out in a pediatric population, and also the evaluation of snoring was subjectively assessed by quality-of-life questionnaires, subject to the parent’s evaluation.

Baxter et al. [40] published, in 2020, a prospective study on the consequences of frenotomy in combination with MFT, finding functional improvements in speech, feeding, and sleep. Even though the results were promising, there is the need to consider that the assessment relied on parental reports. Although it can be considered reliable and feasible, we believe that objective sleep studies are necessary at the moment of comparison between different interventions and investigations. As with nasal breathing, the tongue position is important for a patent airway, and ankyloglossia modifies the UA anatomy and breathing.

In the lecture given by Professor Dr. Eric Kezirian in 2020, available on his Youtube^®^ Channel [48], this referent in OSA noticed that ankyloglossia surgery was not an OSA treatment, and also has never been shown to affect tongue position, as no objective studies were addressing the consequences of this procedure. Despite the fact that frenotomy is not considered as an isolated treatment for OSA, it may have a role as an adjuvant for MFT, whose benefits have been demonstrated [27], to improve compliance and outcomes of oropharyngeal exercises. Furthermore, this is the first study to evaluate the effects of a frenotomy in the upper airway of patients with OSA objectively.

A case report published by Jaikumar et al. [43] in two patients, 14 and 20 years old, showed the benefits of a combination of MFT and frenotomy. Immediately after surgery, the tongue mobility was improved, but limited. It was with the practice of tongue exercises that a significant improvement was recorded.

This concept is also supported by Bargiel et al. [42]. They performed frenotomy in six patients, who were performing speech therapy before the procedure, and continued with tongue exercises in the immediate postoperative period, followed by MFT. Even though they published a step-by-step surgical technique, a so-called miofrenuloplasty, they also concluded that a predictable result requires the cooperation of a speech therapist. 

However, to this date, there are no studies objectively comparing the preoperative and postoperative outcomes of a frenotomy in OSA patients. Our cases have shown that a frenotomy is able to modify the tongue’s position and thus help to prevent its collapse into the airway in two of three patients. The third one showed no changes after the procedure. These findings are the opposite of what was expected, that untethering the tongue would produce glossoptosis with a worsening of the UA collapse. As far as we are aware, this is the first study to use DISE to monitor the UA pattern of collapse in patients with OSA and ankyloglossia before and after frenotomy.

In our cases, we have identified a change in the tongue level (T level in the VOTE classification [36]) with an improvement in the pattern of airway collapse after frenotomy in two of the three patients. How this improvement occurs is not clear, but we demonstrated that untethering the tongue, in a patient with a standard or adequate muscular strength, allows it to take its place in the oral cavity, thus avoiding airway collapse at this level. To explain this effect, we must consider that the tongue is not a rigid structure, but a muscular one with certain adaptative characteristics. In patients with a normal lingual frenulum and its mobility, the tongue contacts the hard palate, adopting in its dynamic form a configuration where the retroglossal space is not invaded (Figure 2). In patients with ankyloglossia, the tongue is tied anteriorly to the floor of the mouth, and it cannot reach the hard palate, so it adapts its form by increasing its posterior volume, invading the space of the oropharyngeal airway (Figure 3).

It was hypothesized originally that untethering the tongue would provoke glossoptosis with an immediate airway collapse. Following this line of thought, if just the tethering of the tongue would be able to prevent its collapse, then we should use a simple suture technique to prevent a tongue base collapse in OSA, avoiding the increased morbidity of current techniques such as Coblator^®^ [49,50,51] or TransOral Robotic Surgery [52,53,54].

On the contrary, in these selected patients, undertaking MFT (i.e., the lingual muscular strength was within normal parameters), releasing the tongue did not result in glossoptosis. We believe this was because of the prior efforts at improving muscular strength, so releasing the tongue helped it to take its optimal position in the oral cavity. Without restraint, the exercised tongue can work in the usual way (Figure 4).

The referred above study by Bussi et al. [41] described that, after frenotomy, there was an improvement in sleep quality due to the tongue “…being able to rest on the palate instead of resting in the floor of the mandible”. This conclusion supports our explanation of the improvement we observed here, as it was also expressed by Baxter et al. [40] that “a tongue-tie prevents the tongue from resting on the palate”.

Although the relationship between ankyloglossia and OSA is still under discussion, more evidence has been published supporting this association. In 2016, Guilleminault et al. [11] showed that a short lingual frenulum is associated with OSA. The higher risk for OSA in patients with ankyloglossia has been supported by the findings of Pia Villa et al. in 2020 [55] and the above cited literature review conducted by Bussi [41]. According to the literature correlating ankyloglossia and OSA, we currently perform a routine evaluation of the lingual frenulum and mobility in every patient [16,26,34], despite most sleep society airway evaluation guidelines do not recommend it [56].

In our experience, every patient with ankyloglossia requires mandatory MFT and speech therapy, with the objective of improving the musculature of the tongue; otherwise, it will fall into the UA when a frenotomy is performed. This recommendation is supported by several authors such as Murias et al. [57], who published a systematic review regarding the laser technique concluding that it “does not eliminate the need for myofunctional exercises and work with a speech therapist”, and Pia Villa et al. [58], who showed that their patients with OSA had poor tongue strength and benefited from MFT.

Following the current concept of performing nasal procedures (e.g., turbinoplasty and septoplasty), not for nasal benefit directly but to improve adaptation and the adherence to CPAP therapy, we consider that a frenotomy may be an effective and necessary procedure for those patients performing MFT, to improve compliance, as we have demonstrated previously [27], and the outcomes of oropharyngeal exercises.

Thus, it is important to evaluate lingual frenulum and tongue motility in the clinical practice, especially in the OSA field. As sleep surgeons, we are not presented with many patients with ankyloglossia, so it is difficult to perform a more extensive population study. However, we hope to encourage other groups to perform the described three-step procedure, as further research in larger populations will confirm or invalidate our findings.

## 5. Conclusions

The results of this study suggest that a frenotomy in OSA patients with ankyloglossia could reduce tongue collapse, probably by allowing the tongue to take its optimal position in the oral cavity. 

These patients should undergo speech therapy and oropharyngeal exercises prior to any surgical procedure, in order to avoid glossoptosis and improve their quality of life and sleep apnea results.

## Figures and Tables

**Figure 1 jcm-12-00201-f001:**
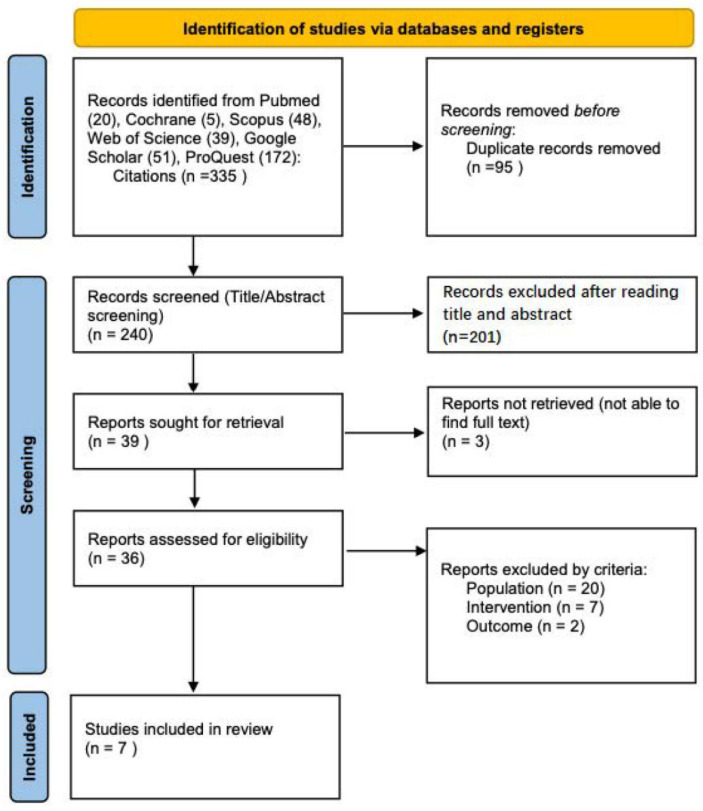
Systematic review identification and selection of studies on ankyloglossia and sleep apnea.

**Figure 2 jcm-12-00201-f002:**
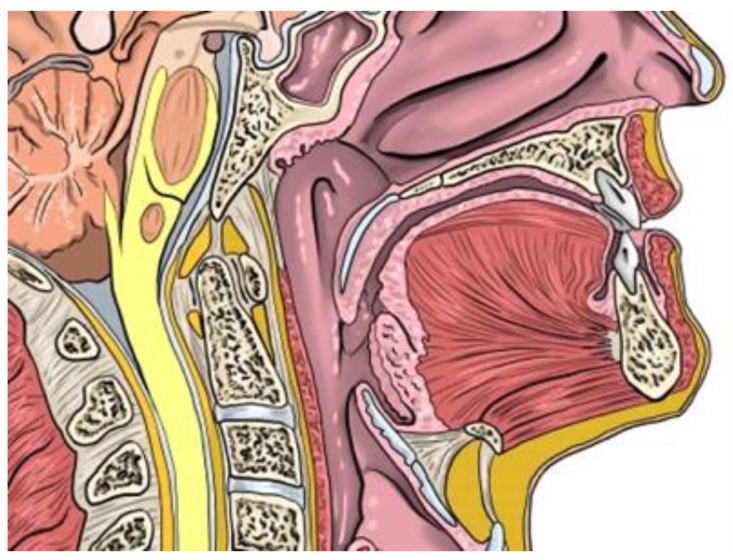
Normal position of the tongue.

**Figure 3 jcm-12-00201-f003:**
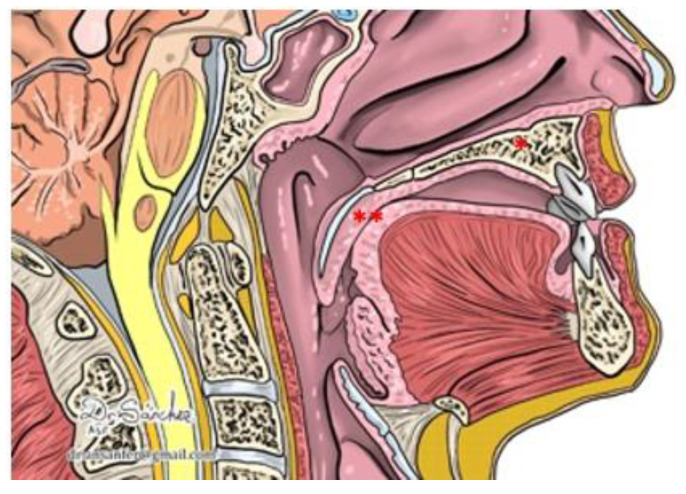
Ankyloglossia: The tongue is not in contact with the hard palate (*) and is invading the retroglossal space (**).

**Figure 4 jcm-12-00201-f004:**
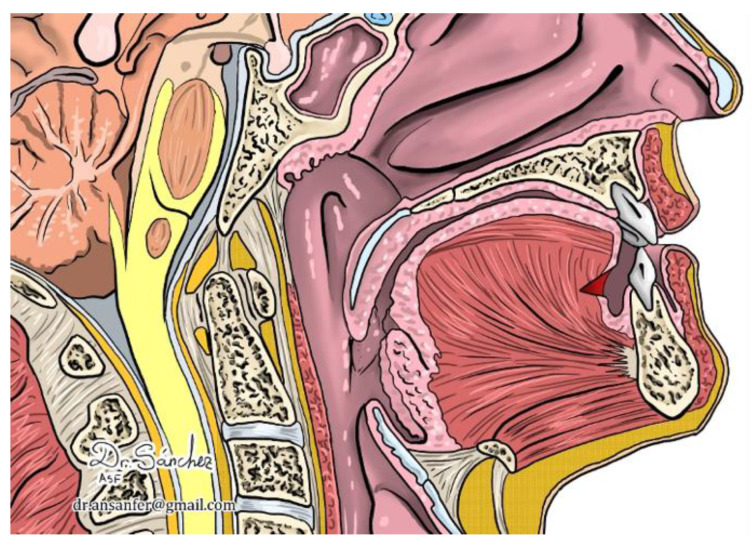
After frenotomy, the tongue takes a normal position.

**Table 1 jcm-12-00201-t001:** Summary of systematic review on OSA and tongue-tie.

Author		Year	Study	Age	Intervention/Exposition	Outcome	Assessment	Conclusion
Guilleminault et al.	[11]	2016	Retrospective	3 to 12 years	Correlation ankyloglossia and obstructive sleep apnea	Correlation of ankyloglossia with sleep apnea	Sleep questionnaires and polysomnography in a sleep laboratory	A short lingual frenulum left untreated at birth is associated with obstructive sleep apnea at later stage.
Zaghi et al.	[39]	2019	Retrospective cohort	29 months to 79 years (*n* = 348)	MFT + frenuloplasty	Mouth breathing, snoring, clenching, myofascial tension	QoL Likert scale—like questionnaires	Safe and potentially effective for mouth breathing, snoring, clenching and myofascial tension.
Messner et al.	[5]	2020	Consensus statement	Children	Case reports	Correlation of ankyloglossia with sleep apnea	Consensus	There is no evidence that ankyloglossia causes sleep apnea.
Baxter	[40]	2020	Prospective	13 months to 13 years	MFT + frenectomy	Speech, feeding, sleep	QoL Likert scale—like questionnaires	Combination of MFT with frenectomy shows functional improvements in speech, feeding, and sleep.
Bussi et al.	[41]	2021	Systematic review	Children	Is ankyloglossia associated with obstructive sleep apnea?	An untreated shortened lingual frenulum at birth is associated with sleep apnea.	Systematic review	Lingual frenuloplasty associated with myofunctional therapy is effective in the treatment of snoring.
Bargiel et al.	[42]	2021	Prospective	18 to 37 years	MFT + miofrenuloplasty	Tongue mobility, swallowing, speech, snoring, neck muscle tension	Tongue mobility (Tongue range motion ratio) and patient referred symptom report	Miofrenuloplasty is and advanced, but effective and highly predictable procedure.
Jaikumar et al.	[43]	2022	Case report	14 and 20 year old	MFT + laser-assisted frenectomy	Tongue mobility	Tongue mobility	Combination of MFT with frenectomy shows benefits and better results.

## Data Availability

Not applicable.

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
