# Peer review of "Does Frenotomy Modify Upper Airway Collapse in OSA Adult Patients? Case Report and Systematic Review"

_jcm, 2022, doi:10.3390/jcm12010201_

Round 1

Reviewer 1 Report

This is a crucial study emphasizing the importance of a lingual frenulum evaluation in adult sleep apnea (OSA) patients. In our daily practice, we may face OSA patients who show the restriction of normal tongue movement caused by ankyloglossia. Unfortunately, this is often forgotten. Therefore, the topic of research undertaken by the authors is of great importance.

As the authors refer to the study by Yoon AJ, restricted tongue mobility can be associated with the narrowing of the maxillary arch and elongation of the soft palate increasing the risk for OSA.

Due to my responsibilities as a reviewer, I'd like to point out some suggestions to improve the manuscript and changes to make.

Title:

1. Consider adding adult patients in the title.

Abstract:

1. Consider changing ap into A-P.

2. Line 32: Change "its optimal position" into the physiological position.

3. Line 33: Chnage "These patients should perform...." into These patients should undergo ...."

Introduction

1. Line 41: Add pediatric population. References 1-4 are related to children.

2. Line 42: Add a dot at the end of the sentence.

3. Line 55: Change reference {16} into [13,14].

4. Line 58: Add references as the authors are saying "....different authors" or rewrite that sentence.

There is a need to add a sentence about treatment options, e.g. frenotomy, frenectomy, MFT.

Materials and methods:

1. Line 87: explain the abbreviation MeSH as Medical Subject Readings.

2. Line 95: Add were (Inclusion criteria were....)

3. Line 98: Add were (Exclusion criteria were...

Case Report:

1. Please explain shortly the method of the examination of the frenulum. The authors referenced "Tongue+protocol" however there is a need to add their way of frenulum examination, e.g. ask patients to elevate the tongue up, etc. Add whether the measurement of tongue mobility was performed.

2. Line 122-123 move up to line 116.

3. Line 172, 176, 178: Add male

4. Line 183,184: Consider changing ap into A-P.

Discussion

1. Line 237: Reference 46 is related to Friedman, not Kezirian.

2. There is a need to mention the importance of nasal breathing for the physiological position of the tongue which is on the hard palate.

3. Fig 3: Mark a hard palate, retroglossal space, and a tongue with arrows.

4. Line 305: Add the reference number by Bussi> I believe it should be 43.

5. Line 316: Delete 38

Conclusion:

1. Line 327: Change into: These patients should undergo...

References:

Please follow the instructions for authors as in some references first and last names are in diffrent positions, for exmple reference 11 and 13.

Author Response

First of all, let us thank you for your time and effort while reviewing our manuscript.

Here you have attached our answers.

Reviewer 2 Report

1. The topic regarding "does frenotomy modify upper airway collapse in OSA patients" is quite interesting and may trigger a heated discussion like oropharyngeal myofunctional therapy.

2. Major comments: 

(1) The reviewed articles were miscellaneous that involved retrospective study, case report, consensus statement and systemic review. The outcome approach was also variable including relationship between ankyloglossal and OSA, mouth breathing, combined MFT and frenectomy. The aforementioned composition makes this paper loosely organized with no main axis.

(2) The use of DISE to assess the changes of upper airway following frenotomy is attractive. The results showed two improved and one remained in terms of tongue collapse. However, the sample size was too small to draw any conclusion, not to mention confounding factors.

(3) The hypothesis for improvement of tongue collapse following frenotomy used in this article is "frenotomy allows the tongue to take its place in the oral cavity". This explanation could be ambiguous. Frenotomy could modify tongue position to stay with upper teeth that ameliorates mouth breathing and improves tongue collapse.

(4) The role of frenotomy and oropharyngeal exercise needs to be justified. A meta-analysis of this issue may help elucidate the controversy and is of great help to the readership.

Author Response

(The authors gave the same response as above.)

Round 2

Reviewer 2 Report

Title:…… in Adult OSA Patients?

The authors responded the comments well

No more questions